# Physiotherapy-Led Health Promotion Strategies for People with or at Risk of Cardiovascular Diseases: A Scoping Review

**DOI:** 10.3390/ijerph20227073

**Published:** 2023-11-16

**Authors:** Etienne Ngeh Ngeh, Anna Lowe, Carol Garcia, Sionnadh McLean

**Affiliations:** 1Research Organization for Health Education and Rehabilitation-Cameroon (ROHER-CAM), Mankon, Bamenda P.O. Box 818, Cameroon; 2Department of Allied Health Professions, Sheffield Hallam University, L108, 36 Collegiate Crescent, Sheffield S10 2BP, UK; a.lowe@shu.ac.uk (A.L.); c.m.garcia@shu.ac.uk (C.G.); s.mclean@shu.ac.uk (S.M.)

**Keywords:** physiotherapy, health promotion, cardiovascular diseases, risk factors, interventions

## Abstract

Cardiovascular diseases (CVD) are prevalent and lead to high morbidity and mortality globally. Physiotherapists regularly interact with patients with or at risk of CVDs (pwCVDs). This study aimed to assess the nature of existing evidence, interventional approaches used, and the population groups included in physiotherapy-led health promotion (PLHP) for pwCVDs. The scoping review followed the Preferred Reporting Items for Systematic Reviews and Meta-Analyses Extension for Scoping Reviews (PRISMA-ScR) guidelines. Medline, PubMed, Web of Science, Cochrane Central Register of Controlled Trials, CINAHL, and PEDro databases were searched from inception until June 2023. Two reviewers independently screened the titles, abstracts, and full text and conducted data extraction. All conflicts were resolved with a third reviewer. A total of 4992 records were identified, of which 20 full-text articles were included in the review. The studies had varied populations, including those with stroke, coronary artery diseases, peripheral artery diseases, hypertension, diabetes, and multiple CVD risk factors. The interventions ranged from exercise and physical activity programmes, dietary interventions, education, and counselling sessions with various supplementary approaches. Most interventions were short-term, with less than 12 months of follow-up. Interventions were personalised and patient-centred to promote adherence and health behaviour change. Among the included studies, 60% employed experimental designs, with the remainder using quasi-experimental designs. Although a wide range of PLHP strategies have been used for pwCVDs, exercise and physical activity were employed in 85% of the included studies. Other components of health promotion, such as sleep, smoking, and alcohol abuse, should be investigated within PLHP.

## 1. Introduction

Cardiovascular disease (CVD) includes all diseases that affect the cardiovascular system (the heart and blood vessels). CVD is the leading cause of disability and mortality globally [1] and accounts for the highest proportion (44%) of chronic non-communicable disease deaths annually [2] and 32% of all deaths globally [3,4]. CVDs are associated with severe social consequences, including reduced quality of life and economic growth, and consume many health service resources in developing and developed countries [5,6].

Annually, approximately 15 million people globally suffer a stroke [7], estimated to rise to 77 million by 2030 [8]. The cumulative risk of stroke recurrence at five years is 1.3%, and at ten years is 39.2% [9], with a higher death and disability rate associated with recurrent stroke [10]. Chronic non-communicable diseases are associated with multiple risk factors, including genetic and environmental factors, metabolic factors (hypertension, diabetes, abnormal lipids, obesity), and behavioural factors (tobacco use, unhealthy diet, physical inactivity) [11]. These risk factors are drivers of the global CVD epidemic [6,12]. A global study in 52 countries identified similar risk factors for heart diseases in low- and high-income countries [12]. However, over three-quarters of the global burden of CVD is from low- and middle-income countries (LMICs), with a rising incidence [13,14].

The effectiveness of lifestyle changes and physical interventions is well established in the primary, secondary, and tertiary prevention of CVDs [15,16]. Primary prevention aims to reduce the incidence of an index cardiovascular event, especially in at-risk people [11,17]. Secondary and tertiary prevention programmes are often provided at specialised cardiac rehabilitation centres and directed towards reducing the recurrence of cardiac events, restoring patients’ quality of life, improving functional capacity, stress, and self-management techniques, and promoting a healthy lifestyle [18]. There is evidence that cardiovascular mortality can be reduced and signs and symptoms of established CVD improved by addressing behavioural risk factors such as an unhealthy diet [19], physical inactivity [20], harmful use of alcohol [21], tobacco use [22], inadequate sleep [23], and poor stress management [3,24,25]. Adopting these preventive strategies may reduce the incidence of heart disease [26,27].

Physiotherapists play a role in reducing risk and managing patients at risk or with established CVD (pwCVDs) [28,29,30]. Despite the substantial burden of CVDs and the evidence supporting cardiac rehabilitation in preventing and managing CVDs, many LMICs do not have existing structures and programmes promoting the prevention and rehabilitation of pwCVDs [18,31,32]. Cardiac rehabilitation services are available in 80% of European countries, but only 17% of African countries [31]. It is therefore important to scale up rehabilitation for pwCVDs in accordance with the WHO call for action “Rehabilitation 20230 [33]. Cardiac rehabilitation services are rare in LMICs for several reasons, including lack of personnel resources, competing priorities, affordability issues, and insurance coverage [32]. Physiotherapists in LMIC settings receive pwCVDs in their practice, providing an opportunity to provide cardiac rehabilitation-related interventions through health prevention and promotion. The contact time and frequent visits make them well-placed to provide physiotherapy-led health promotion (PLHP). PLHP refers to the approach within the field of physiotherapy that focuses on promoting overall health and well-being through education, lifestyle modification, and preventive strategies beyond acute care [34]. Both promotive and preventive strategies, such as health education and the use of exercise in disease prevention and management, are at the core of physiotherapy practice. Given the rising incidence of CVDs and the lack of cardiac rehabilitation services in LMICs, it is essential that physiotherapists from these countries are able to deliver health promotion strategies effectively given the lack of advanced treatment opportunities for these patients. However, no evidence exists to inform or enhance PLHP practice globally.

Previous reviews on PLHP are limited and focused on health education strategies for lifestyle-related conditions in general [35], promoting physical activities [28,36], entry-level training and physical activity promotion [36], and physical activity in cystic fibrosis patients [37]. There are no reviews investigating PLHP strategies for pwCVDs. Consequently, a review is warranted to systematically scope and map out the existing evidence in this area. This review summarises the available literature with the following objectives:To assess the characteristics of existing evidence on PLHP for pwCVDs globally.To identify the interventional approaches that have been used in PLHP strategies for pwCVDs.To evaluate the type of population groups included in the PLHP research.

## 2. Methodology

A scoping review was used to identify and synthesise data on PHLP strategies and interventions in the literature and map existing evidence’s characteristics without critically appraising the methodological quality [38,39]. The methodological framework published by Arksey and O’Malley and the methodological advancement by Levac and colleagues were adopted for this study [40,41]. The Preferred Reporting Items for Systematic Reviews and Meta-Analysis Extension for Scoping Reviews (PRISMA-ScR) recommendations were used for reporting this systematic scoping review [42,43]. The proposed stages in this framework are: (1) Identifying the research questions, (2) identifying relevant studies, (3) study selection, (4) charting the data, and (5) collating, summarising and reporting [41]. The template for intervention description and replication (TIDieR) framework was used to extract intervention data from the included studies. The study protocol was registered on the Open Science Frame (OSF) (OSF.IO/BFZ6Y). This review involved no direct contact with patients or healthcare professionals but reviewed and synthesised already-published data, and therefore was not subject to ethical approval.

### 2.1. Identifying the Research Question

Scoping review questions are generally broad and aim to summarise the available evidence of interest [43]. Based on the overall project aims, the following questions were identified for the present study: (1) What are the characteristics of existing evidence of PLHP for pwCVDs globally? (2) What interventional approaches have been used in PLHP strategies for pwCVDs globally? (3) What population groups have been included in the PLHP research globally?

### 2.2. Identifying Relevant Studies (Database and Search Strategy)

The following electronic databases, registries, and search engines were searched for eligible articles from the inception of the database to June 2023: MEDLINE, PubMed Web of Science, Cochrane Central Register of Controlled Trials, EMBASE, CINAHL, PEDro, Google Scholar, the EU clinical trial register, African Index Medicus, World Physiotherapy Conference proceedings, trials registries, and the World Health Organisation International Clinical Trials Registry Platform portal. A search strategy that considered relevant index terms and keywords was developed with assistance from an experienced librarian from Sheffield Hallam University (Table 1). A subject librarian at Sheffield Hallam University further reviewed this. The search strategy for MEDLINE (final) was adapted for searches in other included databases. Search filters such as publication in the English language, human species, and primary studies were used in relevant databases. References to identified previous and adjacent reviews and included papers were also screened. A complete MEDLINE search strategy can be found in Appendix A.

### 2.3. Eligibility Criteria

Studies were included if they reported or evaluated health promotion for pwCVDs, were led by physiotherapists, and were published in English. Studies with a focus on specific clinical or therapeutic outcomes rather than health promotion were excluded. Details on inclusion and exclusion criteria are provided in Table 2.

### 2.4. Study Selection (Screening)

Studies identified through searches were imported to Covidence, and duplicates were removed. Two independent reviewers (ENN, CG) individually screened studies using a three-step process: First titles, then abstracts (Cohen’s kappa score = 0.43), and finally, full text was screened based on the inclusion and exclusion criteria (Table 2). The full texts of selected studies were reviewed in detail against the inclusion criteria by two independent reviewers (Cohen’s kappa score = 0.35) (ENN, SM). All reasons for excluding potential studies that did not meet the inclusion criteria are reported on the PRISMA flowchart. Any disagreements between the reviewers at each stage of the study selection process were resolved through discussion, and where an agreement was not met, a third reviewer (AL) was consulted.

### 2.5. Data Charting (Data Extraction)

Data charting is the method for extracting data for scoping reviews [40,42]. The chart included information about study participants and the design. Data about the nature of the intervention(s) were extracted based on the TIDieR framework, including the theoretical framework (why), intervention type (what), intervention duration (when), intervention provider (who), delivery format (how), intervention location (where), number of intervention sessions (how much), personalised intervention (tailoring), and fidelity (how well). Intervention duration of less than 12 months was described as short and more than 12 months as long. Total intervention sessions less than 15 sessions and 16 sessions and above were described as low and high volume, respectively. In cases of missing data or insufficiently described processes, the corresponding authors were contacted to clarify or provide the missing information. Screening and data extraction were completed in Covidence.

### 2.6. Quality Appraisal

Based on current guidance for conducting scoping reviews, quality appraisal was not considered necessary to achieve the aims of this study [40,43].

### 2.7. Consultation

We consulted relevant stakeholders, experts in the field, and key informants in the later stages of this review to clarify missing information, identify relevant studies that are ongoing, or identify interventions/concepts not considered in the review [44].

### 2.8. Collating, Summarising, and Reporting

Results are synthesised narratively and presented in a table format based on elements of the TIDier framework.

## 3. Results

### 3.1. Literature Search and Included Studies

The PRISMA flow chart (Figure 1) summarises search results and the methodological steps to arrive at the included studies. The search yielded 4381 articles with the respective numbers for each database, as shown on the PRISMA flow chart (Figure 1). After removing the duplicates, 1716 studies remained and were screened for eligibility. After screening titles and abstracts, 227 articles were deemed potentially eligible. Following full-text screening, 20 studies were included in this review. Reasons for exclusions are documented on the PRISMA flow chart.

### 3.2. Characteristics of Included Studies

Table 3 summarises the characteristics of the included studies. Of the twenty included studies, 12 were randomised controlled trials (RCTs) [45,46,47,48,49,50,51,52,53,54,55,56], seven were quasi-experimental studies [57,58,59,60,61,62,63], and one was a secondary analysis of trial data. All included studies were published between 2002 and 2022. Thirty percent of the included studies were published between 2016 and 2020 [55,57,58,59,64,65]. Studies were conducted on patients with stroke (*n* = 4) [53,57,58,59], risk factors for CVD (*n* = 4) [47,50,54,62], coronary heart diseases (*n* = 3) [51,52,63], peripheral arterial diseases (*n* = 2) [45,56], diabetes (*n* = 3) [48,52,60], weight/obesity (*n* = 2) [49,61], and hypertension (*n* = 2) [46,55]. The sample size of the included studies ranged from 18 to 882 participants [45,46]. The included studies were conducted in 15 countries, with England [56,61], Australia [45,57], Spain [46,63], The Netherlands [48,49], and Norway [58,64] having two studies each and the remaining countries having one study each (Table 4). The majority of studies were from high-income countries (HICs), with 47% from Europe alone, and only two publications [54,55] from two LMICs (Brazil and China). No studies were identified from the African continent.

### 3.3. Characteristics of the Included Interventions

All studies were either solely implemented by physiotherapists (*n*=13) [45,46,48,50,52,53,54,55,56,57,58,59,61] or in combination with other healthcare professionals (nurses, physicians, and dieticians/nutritionists), with physiotherapists leading defined components of the intervention (*n* = 7) [47,49,54,60,62,63,65]. The identified interventions were heterogeneous and reported according to the TIDieR framework in Table 4. 85% of studies used multimodal intervention strategies, with only 15% of studies using a single intervention strategy [46,51,56]. Seven (35%) of the 20 publications employed behaviour change approaches and psychological models such as the theory of planned behaviour and the common-sense model of illness representations (*n* = 1) [56], health belief model and transtheoretical model to promote participant exercise behaviours (*n* = 1) [50], Bandura’s self-efficacy theory (*n* = 1) [63], and motivational interviewing (*n* = 4) [51,56,58,62]. The majority of interventions included exercise or physical activity (*n* = 18), education on lifestyle (*n* = 2) [52,59], and dietary education in combination with another physiotherapists’ led intervention (*n* = 6) [47,49,50,52,55,63]. Seven studies employed behaviour change programmes focused on physical activity uptake (*n* = 5) [51,56,57,61,63] and diabetes management (*n* = 2) [55,60] with only two underpinned by behaviour change theory [56,63]. Self-management and home programmes were also identified (*n* = 4) [48,52,57,62]. Individualised coaching on physical activity and exercise (*n* = 5) [49,58,60,62,64] and use of the health improvement card (HIC) (*n* = 1) [54] were also used by physiotherapists to enhance activity and reduce cardiovascular risks, with only two studies (*n* = 2) reporting employing behavioural change techniques [62,64]. Six studies were characterised by the provision of educational materials/resources, including brochures on healthy lifestyle practices and lifestyle behaviour change (*n* = 2) [45,54], written instructions and recommendations (*n* = 2) [47,59], workbooks (*n* = 1) [55], and handouts following each session (*n* = 1) [61]. Technology-based strategies were also used to deliver interventions for weight management (video-conferencing sessions with real-time communications and the use of remote monitoring using Fitbit) (*n* = 1) [60], video/television programme called Sit and Be Fit during the exercise phase (*n* = 1) [62], and videos on specific exercises and techniques (*n* = 1) [50]). Six studies were supplemented by telephone calls (*n* = 6) [48,50,51,52,56,58]. Adherence to interventions was reported in nine studies [49,50,56,57,58,59,60,62,63].

**Table 3 ijerph-20-07073-t003:** Components and characteristics of the interventions in the included studies for pwCVDs.

	Overall Aim of Intervention	Education on Lifestyle	Dietary Education and Physiotherapy	Exercise and/or Physical Activity	Self-Management and Home Programmes	Behaviour Change Programmes on Physical Activity Uptake	Individualised Coaching on Physical Activity and Exercise	Health Improvement Card (HIC)	Provision of Educational Materials/Resources Such as Brochures on Healthy Lifestyle Practices and Lifestyle Behaviour Change	Workbook	Written Instructions and Recommendations	Handouts Following Each Session	Technology Based	Theory-Based Intervention	Supplemented by Telephone Calls
Fowler et al., 2002 [45]	Improving maximum walking distance in early peripheral arterial disease			**✓**					**✓**		**✓**				
Bonet et al., 2003 [46]	Evaluate, in women with grade 1 essential hypertension, the response of cardio-respiratory and blood pressure after 6 weeks of supervised physical exercise vs. only recommended exercise			**✓**							**✓**				
Eriksson et al., 2006 [47]	Lifestyle intervention programme in primary healthcare		**✓**	**✓**					**✓**						
Quinn et al., 2008 [61]	The effect of a physical activity group-based education programme on weight reduction, physical activity, cardiovascular fitness, quality of life	**✓**		**✓**		**✓**						**✓**	**✓**		
Pariser et al., 2010 [60]	Active Steps for Diabetes			**✓**		**✓**	**✓**				**✓**				
Wisse et al., 2010 [48]	Assess the impact of personalized exercise prescription on habitual physical activity and glycemic control in sedentary, insulin treated type 2 diabetes patients			**✓**	**✓**						**✓**	**✓**			**✓**
Molenaar et al., 2010 [49]	Nutritional counselling and nutritional plus exercise counselling in overweight adults		**✓**	**✓**			**✓**								**✓**
Wu et al., 2011 [50]	Evaluate short- and long-term effects of home-based exercise on adiponectin, exercise behaviour and metabolic risk factors in middle-aged adults at diabetic risk		**✓**	**✓**										**✓**	
Reid et al., 2012 [51]	Evaluate long-term physical activity levels between a theoretically guided motivational counselling (MC) intervention group and a usual care	**✓**		**✓**		**✓**							**✓**	**✓**	**✓**
Oerkild et al., 2012 [52]	Home-based cardiac rehabilitation	**✓**	**✓**		**✓**										**✓**
Takatori et al., 2012 [53]	Investigate the effect of intensive rehabilitation on physical and arterial function among community-dwelling stroke survivors	**✓**		**✓**											
Preston et al., 2016 [57]	Improving self-management				**✓**	**✓**									
Higgs et al., 2016 [59]	Acceptability of a community-based lifestyle programme for adults with diabetes/prediabetes	**✓**		**✓**			**✓**								
Gunnes et al., 2018 [58]	To investigate adherence to an 18-month physical activity and exercise programme			**✓**			**✓**							**✓**	**✓**
Gunnes et al., 2019 [64]	To assess the associations between participants’ degree of adherence to physical activity and exercise and motor function 18 months after inclusion	**✓**		**✓**			**✓**								
Bai et al., 2020 [54]	Health improvement card (HIC) on lifestyle practices and biometric variables in community-dwelling Chinese participants			**✓**				**✓**	**✓**						
Gerage et al., 2020 [55]	To investigate the efficacy of a behaviour change programme on cardiovascular parameters in hypertensive patients		**✓**	**✓**		**✓**				**✓**					
Batsis et al., 2021 [62]	Technology-based weight management intervention for rural older adults with obesity			**✓**			**✓**					**✓**		**✓**	
Bearne et al., 2022 [56]	The effect of a home-based, walking exercise behaviour change intervention in adults with peripheral arterial disease and intermittent claudication			**✓**		**✓**									**✓**
Deka et al., 2022 [63]	The effectiveness of a dietary-education and high-intensity interval resistance training programme on healthy food choices and associated anthropometric variables		**✓**	**✓**		**✓**								**✓**	

✓: Match the corresponding components and characteristics of the interventions with respective studies.

**Table 4 ijerph-20-07073-t004:** TIDIER components and the nature of PLHP interventions for pwCVDs of the included studies.

Author Year	Country	N	Study Design	Population	Nature of the Intervention	Intervention Duration	Theory Use	Mode of, and Delivered by	Setting(s)	Educational Component	Delivery Format	Number of Sessions	Technology	Tailoring	Fidelity
Fowler et al., 2002 [45]	Australia	882	RCT	Males aged 65 to 79 years with PAD	Individual and community intervention, advised participants to walk >30 min daily	Short (12 months)	No	Educational materials and f-t-f by PT	Combined	Yes	Combined	High	No	Yes	No
Bone et al., 2003 [46]	Spain	18	RCT	Overweight women of 30–50 years with grade 1 hypertension	Supervised physical exercise	Short (6 months)	No	Educational materials and f-t-f by PT	Combined	No	Group	High	No	Yes	No
Eriksson et al., 2006 [47]	Sweden	151	Randomised controlled parallel group trial	Patients diagnosed with hypertension, dyslipidaemia, type 2 diabetes, obesity, or any combination thereof are aged 18–65	Lifestyle intervention in primary healthcare	Short (3 months)	No	f-t-f by PT and assistants, dietician and a physician	Clinic	Yes	Group	High	No	Yes	No
Quinn et al., 2007 [61]	Ireland	18	Pre-post-test design	Obese females	Physical activity education	Short (4 months)	No	f-t-f by PT	Clinic	Yes	Individual	Low	No	No	No
Pariser et al., 2010 [60]	USA	22	Pre-post-test design	Type 2 Diabetes patients with impaired mobility issues	Active steps for diabetes (exercise and educational intervention)	Short (2 months)	No	f-t-f by PT (assisted by PT student or nurse/diabetes educator)	Combined	No	Combined	High	Yes	Yes	No
Wisse et al., 2010 [48]	The Netherlands	74	RCT	Sedentary, insulin-treated type 2 diabetes	Regular, structured, and personalised exercise prescription	Long (24 months)	No	f-t-f by PT supplemented with telephone calls	Combined	Yes	Individual	Low	Yes	Yes	No
Molenaar et al., 2010 [49]	The Netherlands	203	RCT	Men and non-pregnant women aged 18–65 years with a BMI of 28–35 kg/m^2^	Nutritional counselling and nutritional plus exercise counselling in overweight adults.	long (13.7 months)	No	f-t-f by Dietician and PT	Clinic	Yes	Individual	Low	No	Yes	Yes
Wu et al., 2011 [50]	Taiwan	135	RCT	People 45 to 64 years old are at risk of developing diabetes	Home-based exercise	Short (6 months)	Yes	f-t-f supplemented with telephone calls by PT	Community	Yes	Individual	High	Yes	Yes	Yes
Reid et al., 2011 [51]	Canada	141	RCT	Patients with acute coronary syndromes	Motivational counselling intervention	Short (12 months)	Yes	f-t-f supplemented by telephone calls	Combined	Yes	Individual	Low	Yes	Yes	Yes
Oerkild et al., 2012 [52]	Denmark	40	RCT	Elderly coronary heart disease above 65 years	Cardiac home programme for the elderly	Short (12 months)	No	home visits in person, follow-up with telephone calls by PT	Community	Yes	Individual	Low	Yes	Yes	No
Takatori et al., 2012 [53]	Japan	44	RCT	Chronic stroke survivors 57–89 years	Exercise therapy	Short (3 monhs)	No	f-t-f by PT	Clinic	No	Individual	High	No	Yes	No
Higgs et al., 2016 [59]	New Zealand	36	Prospective observational	Diabetic or at a high risk of developing diabetes	Education and exercise	Short (3 months)	No	f-t-f by PT, PT students and a nurse.	Clinic	Yes	Individual	High	No	Yes	Yes
Preston et al., 2017 [57]	Australia	20	pre-post-test intervention	Patients with mild to moderate acute stroke	Self-management	Short (3 months)	No	f-t-f by PT	Community	Yes	Individual	Low	No	Yes	Yes
Gunnes et al., 2018 [58]	Norway	186	Prospective longitudinal	Adult stroke patients	Physical activity and exercise programme	Long (18 months)	Yes (MI)	f-t-f and over the phone by PT	Community	Yes	Individual	High	Yes	Yes	Yes
Gunnes et al., 2019 [64]	Norway	186	Secondary analyses of multisite RCT	Stroke patients	Individualised coaching on physical activity and exercise	Long (18 months)	Yes (MI)	f-t-f supplemented by telephone calls by PT	Clinic	Yes	Individual	High	Yes	Yes	Yes
Bai et al., 2020 [54]	China	200	RCT	50–90 years	Health education based on the HIC, individualised exercise programme. Standard brochure on healthy lifestyle practices	Short (3 months)	Yes (HIC)	f-t-f by PT students supervised by PT.	Community	Yes	Individual	Low	No	Yes	No
Gerage et al., 2020 [55]	Brazil	90	RCT	Patients with primary hypertension	Behavioural change programme supplemented with educational materials	Short (3 months)	Yes (VAMOS)	f-t-f by PT	Clinic	Yes	Group	Low	No	No	No
Batsis et al., 2021 [62]	USA	54	Single-arm trial	Older (65+) adults with obesity (BMI > 30 kg/m^2^) residing in rural New Hampshire and Vermont	Technology-based weight management intervention	Short (6 months)	Yes (social cognitive theory, MI)	f-t-f and telemedicine (video conferencing, remote use of Fitbit) and periodic face-to-face interaction onsite. By dietitian and PT	Community	Yes	Combined	High	Yes	Yes	Yes
Bearne et al., 2022 [56]	England	190	RCT	Adults with peripheral arterial disease and intermittent claudication	Walking Exercise Behaviour Change Intervention	Short (6 months)	Yes (theory of planned behaviour and the common sense model of illness representation)	f-t-f and supplemented by telephone calls by PT	Clinic	Yes	Individual	Low	Yes	Yes	Yes
Deka et al., 2022 [63]	Spain	22	Single-arm trial	Patients with coronary artery diseases	Dietary education and a high-intensity interval resistance training programme (DE–HIIRT)	Short (3 months)	Yes (Bandura’ self-efficacy theory)	f-t-f by dietician and PT	Clinic	Yes	Combined	22	No	Yes	Yes

PAD: peripheral arterial diseases, f-t-f: face-to-face, PT: physiotherapist, short: <12 months, long: >13 months, high = 16 sessions, low = <15 sessions, HIC: health improvement card, MI: motivational interviewing, VAMOS: Vida Ativa Melhorando a Saúde, BMI: body mass index, RCT: randomised controlled trial, USA: United States of America.

## 4. Discussion

This review identified the nature of the evidence and the types of interventions used and implemented by physiotherapists for pwCVDs within their scope of practice. This involved opportunistic advice, discussions, encouragement, and strategies that physiotherapists are able to use for disease prevention and health promotion within their profession in addition to their therapeutic role. While health promotion and therapeutic interventions are within the scope of physiotherapy practice, much attention has not been given to physiotherapy health promotion globally. This is the first review explicitly exploring PLHP for pwCVDs globally, providing an opportunity for discussion and future research in this area.

No grey literature was found, and all included studies were published between 2002 and 2022. Given that there were no restrictions in the search period, this is a small volume of literature. This could be explained in two ways. Firstly, the inclusion was based on physiotherapists leading or implementing the intervention, focusing on primary and secondary prevention of CVDs to heart disease risk factors. Based on this criterion, many studies were excluded as not physiotherapist led (*n* = 60) (Figure 1). Secondly, earlier attention to physiotherapists’ interventions was directed towards therapeutic and curative treatment rather than prevention. Over the last two decades, physiotherapy preventive roles have been increasing with the rising burden of CVDS [65,66]. This aligns with the global call for physiotherapists to contribute to preventing lifestyle-related conditions [65,66,67]. The increasing trend in research output indicates that more evidence will emerge in the coming years as physiotherapists gain skills and autonomy in leading prevention programmes.

Currently, most studies have emerged from Europe (55%), with no studies from the African continent. Given the vast burden of CVDs in African countries with unique ethnic, cultural, and context-specific determinants [68,69] and the lack of CR programmes on this continent [31,70], it is essential to see more research investigating PLHP for pwCVDs in African countries to facilitate effective preventive interventions. Only two studies (10%) from LMICs were included in this review, and both were supported with research funding [54,55]. Generally, PLHP research may be difficult to realise in LMIC settings due to a lack of research priorities, funding problems, and a lack of infrastructure and researchers with relevant skills [71,72]. Addressing funding issues by budgeting for the prevention of NCDs in LMICs, among other potential barriers, may contribute positively to data generation for pwCVDs in low-resource settings.

Many of the included studies were RCTs (60%), followed by different quasi-experimental designs (35%). The available data provides an opportunity for follow-up studies, such as a systematic review of effectiveness. This is necessary to determine whether PLHPs are effective for wider-scale adoption. No qualitative work on PLHP was identified, and there is a gap in our understanding of patient perceptions and experiences of PLHP approaches. More research is necessary for designing and implementing PLHP in the future.

Diverse interventional approaches have been used in PLHP for pwCVDs (Table 4). CVD PLHP interventions are likely to be complex, and therefore require a multimodal approach, due to different populations, multiple risk factors for CVD, and non-adherence to recommendations for managing these risk factors [66]. This review included studies focused mainly on exercise and physical activity uptake, weight management, and diet. Other components of health promotion for pwCVDs, such as sleep hygiene, smoking cessation, and alcohol abuse, among others, were not reported. These components are within the scope of physiotherapists, and it is necessary that physiotherapists receive adequate training that can enable them to confidently tackle the multiple risk factors associated with CVD. Qualified physiotherapists should be familiar with assessment tools related to general health measures, lifestyle-related behaviours, and NCD risk factors in general, including how to assess self-efficacy for behaviour change and readiness to change a lifestyle behaviour [65,66]. This should include counselling skills and the use of behaviour change strategies for specific populations. Physiotherapists should work in synergy with other health professions, making appropriate referrals and identifying relevant resources to improve outcomes.

Three studies employed theory-based behaviour change models supported by evidence-based behaviour change techniques such as motivational interviewing to inform and complement their interventions. These behaviour change theories and techniques were adopted in more recent studies published between 2011 and 2022. This indicates an increased understanding of the importance of including behaviour change techniques and theories for effective health education to strengthen patients’ motivation and adherence during and beyond the active rehabilitation period. More rigorous, theoretically informed approaches to support behaviour change for pwCVDs should be included in intervention strategies that facilitate change in lifestyle risk factors. This is also necessary in clinical practice and should be integrated into physiotherapy training [66]. In delivering broad health promotion strategies for pwCVDs, physiotherapists need to receive broader training in addressing these risk factors.

### 4.1. Implications for Clinical Practice

It is sensible to consider PLHP strategies incorporating interventions beyond exercise and physical activity. Understanding and increasing competence in implementing behaviour change in stress management, sleep, nutrition, and weight management through appropriate strategies is necessary for effective PLHP. Dean and colleagues highlighted the need to raise the priority of lifestyle prevention strategies for NCDs [67] and competency standards, including relevant behaviour change approaches [66], to improve practice adequately. Considering the different study populations and the multiple risk factors addressed in the included studies, it is necessary that physiotherapists collaborate with other healthcare providers to optimise health promotion and prevention programmes. Digital and technological monitoring and other interventions have been used successfully in some trials and contexts [60,73]. This can be useful in other contexts while considering local challenges and possible barriers.

### 4.2. Research Implications

The findings of this review demonstrate the lack of studies from Africa and other LMICs, which is concerning given the rising burden of pwCVDs in these regions. For effective interventions to be developed, it is necessary to consider increasing research output in these contexts.

PLHP interventions ought to be multimodal, theoretically informed and supported by behaviour change theories and techniques and delivered by physiotherapists who have been adequately trained and, where necessary, optimised by appropriate health care professionals who complement the physiotherapists’ skills and knowledge. These optimised interventions should also be reported in further trials following the TIDier framework. There is no evidence to characterise the optimal intensity and critical characteristics of weight management programmes for specific populations.

Findings highlight the increased use of digital technology at different levels of PLHP interventions with varied levels of adherence [60,62]. Digital or technological devices that are attractive, affordable, easy to use, and sensitive to specific outcomes in different contexts for better adherence and output should be considered.

Most of the included trials reported short-term follow-up. Despite the prevailing challenges, PLHP interventions are warranted to demonstrate longer-term clinical outcomes.

### 4.3. Strengths and Limitations

Scoping reviews provide breadth and the inclusion of all study designs, which makes this realistic about a topic. This study employed the recommended guidelines for conducting a scoping review with multiple reviewers for data screening and extraction, making the findings rigorous. The broad scope provides a complete overview of PLHP, which has been trialled in primary, secondary, and tertiary health promotion in low- and high-income countries. This provides researchers with clear directions about developing the PLHP based on evidence and where further research needs to be undertaken. Additionally, the results of this scoping review may apply to clinicians employing the identified strategies/approaches as CVDs and their risk factors share and pose similar risks to other CNCDs. This review considers only literature published in English. This might have limited the scope of this review to articles published in or from non-English-speaking countries.

## 5. Conclusions

Based on the literature, physiotherapists are trying to address the growing burden of CVDs through various PLHP strategies. PLHP strategies are focused on exercise and physical activity, and there is a need to tackle CVD beyond addressing sedentary behaviour, considering the multiple risk factors. Assessing the risks and needs, tailoring the interventions to individuals, and monitoring appear central and consistent with practical preventive principles and strategies. It is crucial that physiotherapists work together with other healthcare professionals to optimise relevant components of health promotion effectively. Health behaviour change theories and techniques should be commonly used to support positive health behaviour change, and it may be necessary to provide comprehensive training to integrate lifestyle management approaches in physiotherapy practice. This is even more compelling for physiotherapy practice in Africa and LMICs with huge CVD burdens. Further study is needed to elucidate the effectiveness of existing PLHP interventions for pwCVDs. Avenues for future research have been highlighted.

## Figures and Tables

**Figure 1 ijerph-20-07073-f001:**
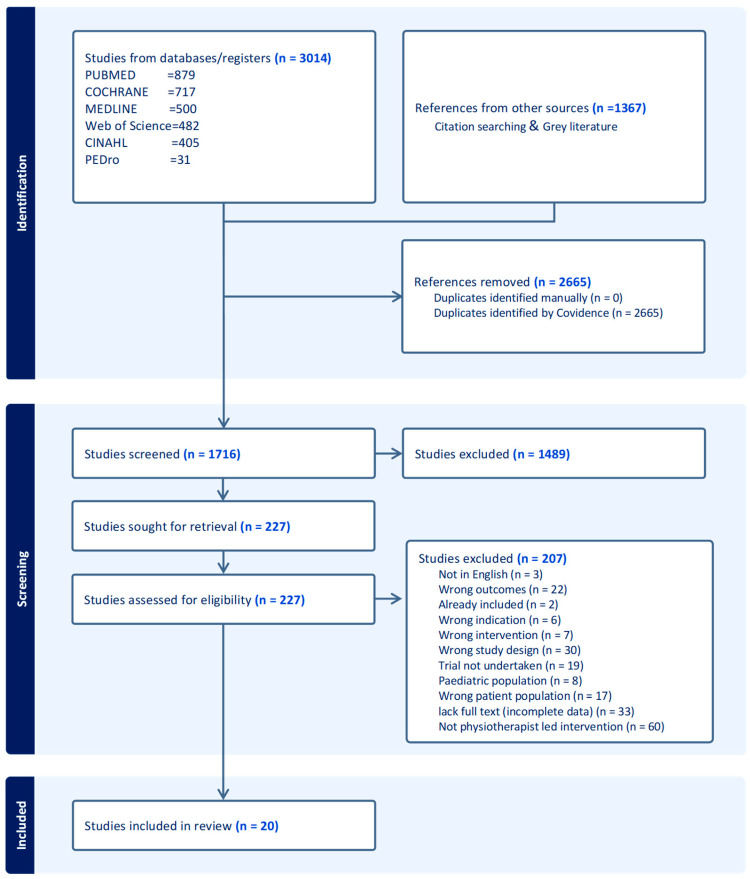
PRISMA flow chart for searches.

**Table 1 ijerph-20-07073-t001:** Search Parameters.

Participants/Population	Concept/Intervention
Cardiovascular disease and risk factors block keywords, cardiovascular diseases, heart diseases, coronary artery disease, coronary heart disease, myocardial infarction, heart failure, angina, cerebrovascular disease, stroke patients, and aortic atherosclerosis patients—overweight, obesity, diabetes, blood pressure, hypertension, dyslipidaemia.	Physiotherapy block keywords: Physiotherapist(s), physiotherapy, kinesiotherapy, physical therapist(s), physiotherapy assistant. Health promotion block keywords: Patient education, health promotion, health education, health behaviour, educational technology, diet therapy, educational health promotion, group-based, individual, home and hospital-based approaches, lifestyle modification, lifestyle change recommendations, physical activity and exercise promotion, brief counselling, face to face, group sessions, skill training, visual presentation, handouts, brochures and diaries, motivational prompts, individualised plan, goal setting, nutrition and weight management, smoking cessation, tobacco exposure, sleep, stress management.

**Table 2 ijerph-20-07073-t002:** Inclusion and exclusion criteria.

Participants/Population	Concept/Intervention	Context	Study Types and Design
Inclusion Criteria			
Patients with CVD risk factors or established CVDs. Studies conducted or implemented by Physiotherapists or Physiotherapy assistants.	Health promotion strategies, including behavioural change, educational, client-centred societal change, etc.	Primary, secondary, and tertiary care settings. Rural and urban settings. Low- and high-income countries.	All eligible primary studies. Both quantitative and qualitative studies published and unpublished articles.
**Exclusion Criteria**			
Studies on pwCVD with relevant outcomes were initiated and implemented by clinicians other than physiotherapists.	Studies on pwCVD but with non-cardiovascular related outcomes. Pure therapeutic intervention with no intention to improve adherence or health behaviours.		(Incomplete research data. letters to the editor, commentaries, notes, reviews. studies in languages other than English. Secondary research with any design will be excluded.

## Data Availability

All data extracted and synthesised in this review were taken directly from the published articles.

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
