# Peer review of "Physiotherapy-Led Health Promotion Strategies for People with or at Risk of Cardiovascular Diseases: A Scoping Review"

_ijerph, 2023, doi:10.3390/ijerph20227073_

Round 1

Reviewer 1 Report

Comments and Suggestions for Authors

Dear authors,

Thank you very much for the effort and diligence you put into your Review research. Your research, in its current form, meets all the scoping review conditions and it is obvious that a lot of effort was put into writing it. I will only give a few suggestions, I recommend you do it if the recommendations I give will not make your review disorganized. If the necessary corrections will disrupt the flow of the article, you may not make the necessary answers. With my most sincere respects.

Table 3 is difficult to read because you only use reference numbers, but if it makes sense for you to arrange it vertically with valuable coverage, please consider it.

In the discussion, you used new references in addition to the references you used in the findings. If there are references from the sources you used in the Results section that you can use in the discussion section, I recommend you evaluate them.

Thank you very much for giving me the opportunity to read this enjoyable article.

With my sincerest regards

Author Response

Dear authors,

Thank you very much for the effort and diligence you put into your Review research. Your research, in its current form, meets all the scoping review conditions and it is obvious that a lot of effort was put into writing it. I will only give a few suggestions; I recommend you do it if the recommendations I give will not make your review disorganized. If the necessary corrections will disrupt the flow of the article, you may not make the necessary answers. With my most sincere respects.

Table 3 is difficult to read because you only use reference numbers, but if it makes sense for you to arrange it vertically with valuable coverage, please consider it.

Authors response; Many thanks for highlighting this. Completed - see new Table 3 on page 9. It has been considered and improved upon.

In the discussion, you used new references in addition to the references you used in the findings. If there are references from the sources you used in the Results section that you can use in the discussion section, I recommend you evaluate them.

Authors response: New references were only used to support discussions beyond the scope of the included articles where necessary

Thank you very much for giving me the opportunity to read this enjoyable article.

With my sincerest regards

Much appreciated.

Reviewer 2 Report

Comments and Suggestions for Authors

1-     Although it might not be the aim of the review to focus on the context of Africa, but given that this point is raised in multiple places through the manuscript, the following literature could be used to further justify this notion.

·      Minja NW, Nakagaayi D, Aliku T, Zhang W, Ssinabulya I, Nabaale J, Amutuhaire W, de Loizaga SR, Ndagire E, Rwebembera J, Okello E, Kayima J. Cardiovascular diseases in Africa in the twenty-first century: Gaps and priorities going forward. Front Cardiovasc Med. 2022 Nov 10;9:1008335. doi: 10.3389/fcvm.2022.1008335. PMID: 36440012; PMCID: PMC9686438.

·      Anand, S., Bradshaw, C. & Prabhakaran, D. Prevention and management of CVD in LMICs: why do ethnicity, culture, and context matter?. BMC Med 18, 7 (2020). https://doi.org/10.1186/s12916-019-1480-9

2-    The authors only referred to the term “smoking cessation” while health promotion does not include smokers only. Non-smokers who are classified as second-hand smokers also suffer from similar consequences to smokers themselves. Especially if the focus is LMICs where regulations of tobacco control are not in action. Therefore, adding other terms like “tobacco exposure” might be better reflective for the context. 

3- The authors did not refer to the health disparities between high and LMICs which are inherently unavoidable topics when discussing differences in health practices especially those including prevention interventions. 

4- The aim of the scoping review lies within the WHO call for action “Rehabilitation 20230”. The call emphasizes the need to scale up rehabilitation by 2030. The authors need to address this somewhere in the Introduction or Discussion sections. 

Corrections: 

·      Page: 2, Line (57, 58)

“Low-and medium income countries LMICs “

Comment: replace the word medium with the word middle (You can check the World Bank terminology)

·      Page: 14, Line (314)

“20xx and 20yy”

Comment: specify the years

·      Page:14, Line (316)

“and education.and to strengthen”

Comment: revise the sentence

·      Page:15, Line (325)

“In effecting behavior”

Comment: do you mean effective?

Author Response

1-     Although it might not be the aim of the review to focus on the context of Africa, but given that this point is raised in multiple places through the manuscript, the following literature could be used to further justify this notion.

  • Minja NW, Nakagaayi D, Aliku T, Zhang W, Ssinabulya I, Nabaale J, Amutuhaire W, de Loizaga SR, Ndagire E, Rwebembera J, Okello E, Kayima J. Cardiovascular diseases in Africa in the twenty-first century: Gaps and priorities going forward. Front Cardiovasc Med. 2022 Nov 10;9:1008335. doi: 10.3389/fcvm.2022.1008335. PMID: 36440012; PMCID: PMC9686438.
  • Anand, S., Bradshaw, C. & Prabhakaran, D. Prevention and management of CVD in LMICs: why do ethnicity, culture, and context matter?. BMC Med 18, 7 (2020). https://doi.org/10.1186/s12916-019-1480-9

Authors response: Sincerely appreciate the articles provided to highlight the context of Africa.  They have been used in the discussion section to highlight the importance of diverse health determinants in context (References 79 & 80, Line 307).

2-    The authors only referred to the term “smoking cessation” while health promotion does not include smokers only. Non-smokers who are classified as second-hand smokers also suffer from similar consequences to smokers themselves. Especially if the focus is LMICs where regulations of tobacco control are not in action. Therefore, adding other terms like “tobacco exposure” might be better reflective for the context.

Authors response: The suggestion to consider smoking beyond smoking cessation was accepted and integrated in Table 1 under section for concept and intervention. See page 4

3- The authors did not refer to the health disparities between high and LMICs which are inherently unavoidable topics when discussing differences in health practices especially those including prevention interventions.

Authors response: Health disparity was highlighted in two instances. First time in the introduction Line 74-83 and in the second instance discussion Line 305-315

4- The aim of the scoping review lies within the WHO call for action “Rehabilitation 20230”. The call emphasizes the need to scale up rehabilitation by 2030. The authors need to address this somewhere in the Introduction or Discussion sections.

Authors response: This has been integrated in the introduction Line 80-82

Corrections:

  • Page: 2, Line (57, 58)

“Low-and medium income countries LMICs “

Comment: replace the word medium with the word middle (You can check the World Bank terminology)

Corrected Line 58

  • Page: 14, Line (314)

“20xx and 20yy”

Comment: specify the years

Corrected line 341

  • Page:14, Line (316)

“and education.and to strengthen”

Comment: revise the sentence

Revised and corrected line 343-344

  • Page:15, Line (325)

“In effecting behavior”

Comment: do you mean effective?

Revised and corrected line 352

Authors response: All corrections were made.

Reviewer 3 Report

Comments and Suggestions for Authors

Thank you for the opportunity to read this paper that focuses on an important area of healthcare and future role for physiotherapists and other health professionals. I support the publication of this paper with the following minor considerations.  

Language: this is well written. There are a lot of abbreviations used in the manuscript and it may be clearer if the authors would consider abbreviating 3-4 frequently used phrases rather than the number currently. 

Introduction: this is a good summary. It may be helpful to more clearly define or indicate how PLHP is different to what might be offered in typical secondary prevention or cardiac rehab programs. It would be good to make it clear if you are referring to health promotion or if you are referring to disease prevention, while these concepts overlap in many areas they are fundamentally different. In my experience physiotherapists are typically involved in disease prevention eg leading disease preventative services and providing information and support on behaviour change at an individual and community level and sits inside healthcare sector. Health promotion (from a true public health lens) really looks at strategies to empower people (groups/communities) to take control of health, determinates of health and increase healthy behaviours, health promotion crosses many sectors, policies and is focused on social determinants of health. This is not clear in the paper, either defined in the introduction or included in the discussion. 

Method: It is interesting that the authors have chosen not to include Arsky and O'Malley's optional stage six: consultation in their design. This would have added to the strength of the paper. A sentence justifying the decision could be helpful. The authors have adhered to the recommendations of PRISMA-ScR and followed the TIDieR framework and the study was registered apriori. The research question is appropriate for a scoping review and the authors have used appropriate databases and appropriate search terms. Please note in the version that I had to review Table 1 and 2 were cut off. I could not read Table 2 past the first column under Study Types and Design. However, Population, Concept, Context and the first Study Types and Design was explained well. 

Results: PRISMA flow chart is clear including the reason for study exclusion at the screening level. In the first paragraph it would be good to include the level of agreement between authors at each stage of the screening with the Kapa scores – should be available through Covidence. Line 211 of results, para 3.2 Characteristics of Included Studies includes (reference these please) – please address this. This is listed a number of times throughout the 3.3 as well. 

To be consistent with your referencing style in the results please add the 2 references for the countries listed on lines 217-8 (England, Australia, Spain, The Netherlands and Norway). 

Table 3 shows a nice message on which interventions were more common, however, it is hard to read on the screen. Can this be made clearer with shading/lines/spacing?

Table 4 – please check for consistency in reporting. Here it might be better to use abbreviations. Eg in row commencing Fowler et al. Males 65-79 years with PAD (and include PAD Peripheral Artery Disease in the key at the end of the table). You have used PAD in the next column. Limit information in the Nature of the Intervention column to the intervention, eg in the row commencing Fowler, you have referred to people with PAD – this would be assumed from the population. For the column Mode of, and Delivered by – what is the difference between Education materials and F-t-f by PT; and f-t-f/supplemented by education materials by PT. If the order is not important it would be better to be consistent in the list provided. In the column Setting(s) – be consistent with Clinic vs clinical setting or make it clear in the text what the difference is. For Delivery Format there are a combination of “group’ and ‘Group’ be consistent with capitalisation of the first letter and there is a combination of ‘Individual’ and ‘Individualised’, it would be better to use one consistently or explain the difference. 

Line 240 – Seven studies employed behaviour change programs … were these programs based on theory or models? Similar line 244 refers to coaching, was the coaching program based on an existing theory or model? It would be good to get a sense on whether the physiotherapists are following models and theory or doing their own thing.

Line 247 refers to education materials – were these developed by the providers or other national/international bodies. Eg in Australia some programs develop their own brochures and patient handouts, whereas others will use information that has been developed by the National Heart Foundation or other National bodies (where there has been some rigor). 

Discussion: 

Avoid repeating results eg first sentence of para 1 and 2. 

I think there needs to be a discussion point around the limited use of health promotion in the papers - the use of health promotion is limited to disease prevention and all interventions sit well within the healthcare sector. There is no evidence in this review that physiotherapists undertake health promotion but rather disease prevention. 

Para commencing “Currently, most studies emerged from Europe…” you mention possible reasons that PLHP of pwCVD is less common in LMICs including funding and researchers will relevant skills. Do you think that physio’s scope of practice enables them to work in health promotion (or disease prevention) in LMICs? There are papers that discuss this issue. 

Line 314 in the paragraph commencing “three studies employed…” there is 20xx and 20yy – this needs to be addressed. 

 It is interesting that the use of technology was so limited to telehealth (video-conferencing) and remote monitoring using a Fitbit, when there have been behaviour change apps being trialled for many years. The search strategy did not intentionally search from digital health intervention as it did not include terms like mHealth, eHealth, telehealth etc which may be why there were limited papers. Also, physiotherapists may not be leading digital interventions because it has not been a part of the entry-level curriculum and we are seeing a lag in uptake of digital strategies. 

Comments on the Quality of English Language

Well written. 

As in previous section, recommend re-considering the use of so many abbreviations. There some abbreviations that could be written in full with little impact on word count and it would improve the readability. 

There are some missing dates and references throughout that need to be addressed. 

Author Response

Thank you for the opportunity to read this paper that focuses on an important area of healthcare and future role for physiotherapists and other health professionals. I support the publication of this paper with the following minor considerations.  

Language: this is well written. There are a lot of abbreviations used in the manuscript and it may be clearer if the authors would consider abbreviating 3-4 frequently used phrases rather than the number currently. 

Authors response: Thank you for highlighting this. Chronic Non-Communicable Disease (CNCD) and Cardiac rehabilitation (CR) have been used through out the text without abbreviation. Other familiar ones and pwCVDs were maintained because of word count.

Introduction: this is a good summary. It may be helpful to more clearly define or indicate how PLHP is different to what might be offered in typical secondary prevention or cardiac rehab programs. It would be good to make it clear if you are referring to health promotion or if you are referring to disease prevention, while these concepts overlap in many areas they are fundamentally different. In my experience physiotherapists are typically involved in disease prevention eg leading disease preventative services and providing information and support on behaviour change at an individual and community level and sits inside healthcare sector. Health promotion (from a true public health lens) really looks at strategies to empower people (groups/communities) to take control of health, determinates of health and increase healthy behaviours, health promotion crosses many sectors, policies and is focused on social determinants of health. This is not clear in the paper, either defined in the introduction or included in the discussion. 

Authors response: Thank you for the pertinent remark and observation. While majority of the strategies identified tend to focus on disease prevention, there is overlap with elements of health promotion which has a wider focus and scope. Attempt have been made to highlight the nature of the interventions as physiotherapist-led (opportunistic) not to be confused with established cardiac rehabilitation interventions in line 286-290

Method: It is interesting that the authors have chosen not to include Arsky and O'Malley's optional stage six: consultation in their design. This would have added to the strength of the paper. A sentence justifying the decision could be helpful. The authors have adhered to the recommendations of PRISMA-ScR and followed the TIDieR framework and the study was registered apriori. The research question is appropriate for a scoping review and the authors have used appropriate databases and appropriate search terms. Please note in the version that I had to review Table 1 and 2 were cut off. I could not read Table 2 past the first column under Study Types and Design. However, Population, Concept, Context and the first Study Types and Design was explained well. 

Authors response: The section on consultation was omitted during the final version but was originally part of the protocol and is now updated Line 200-203.  All tables have now been reformatted or landscaped to improve readability.

Results: PRISMA flow chart is clear including the reason for study exclusion at the screening level. In the first paragraph it would be good to include the level of agreement between authors at each stage of the screening with the Kapa scores – should be available through Covidence.

Line 211 of results, para 3.2 Characteristics of Included Studies includes (reference these please) – please address this. This is listed a number of times throughout the 3.3 as well. 

Authors response: Cohens kappa scores have been included in the study selection/screening section Line 170 and 173.

To be consistent with your referencing style in the results please add the 2 references for the countries listed on lines 217-8 (England, Australia, Spain, The Netherlands and Norway). 

Authors response: All the corrections on references have been integrated Line 231-233.

Table 3 shows a nice message on which interventions were more common, however, it is hard to read on the screen. Can this be made clearer with shading/lines/spacing?

Authors response: Table 3 has been reformatted in line with reviewer 2 comments. We hope this improves the readability.

Table 4 – please check for consistency in reporting. Here it might be better to use abbreviations. Eg in row commencing Fowler et al. Males 65-79 years with PAD (and include PAD Peripheral Artery Disease in the key at the end of the table). You have used PAD in the next column. Limit information in the Nature of the Intervention column to the intervention, eg in the row commencing Fowler, you have referred to people with PAD – this would be assumed from the population. For the column Mode of, and Delivered by – what is the difference between Education materials and F-t-f by PT; and f-t-f/supplemented by education materials by PT. If the order is not important it would be better to be consistent in the list provided. In the column Setting(s) – be consistent with Clinic vs clinical setting or make it clear in the text what the difference is. For Delivery Format there are a combination of “group’ and ‘Group’ be consistent with capitalisation of the first letter and there is a combination of ‘Individual’ and ‘Individualised’, it would be better to use one consistently or explain the difference. 

Authors response: All the highlighted corrections on Table 4 have been integrated. Thank you.

Line 240 – Seven studies employed behaviour change programs … were these programs based on theory or models? Similar line 244 refers to coaching, was the coaching program based on an existing theory or model? It would be good to get a sense on whether the physiotherapists are following models and theory or doing their own thing.

Authors response: The behaviour change theories, models or techniques used have been addressed in Line 258 and 263.

Line 247 refers to education materials – were these developed by the providers or other national/international bodies. Eg in Australia some programs develop their own brochures and patient handouts, whereas others will use information that has been developed by the National Heart Foundation or other National bodies (where there has been some rigor). 

Authors response: It is challenging to establish whether educational materials were produced locally or by international body or group as it is not reported.

Discussion: 

Avoid repeating results eg first sentence of para 1 and 2. 

Authors response: This has been addressed. Thank you.

I think there needs to be a discussion point around the limited use of health promotion in the papers - the use of health promotion is limited to disease prevention and all interventions sit well within the healthcare sector. There is no evidence in this review that physiotherapists undertake health promotion but rather disease prevention. 

Authors response: Thank you for highlighting this, it has been considered and highlighted in line 290 -294.    

Para commencing “Currently, most studies emerged from Europe…” you mention possible reasons that PLHP of pwCVD is less common in LMICs including funding and researchers will relevant skills. Do you think that physio’s scope of practice enables them to work in health promotion (or disease prevention) in LMICs? There are papers that discuss this issue. 

Authors response: Thank you for highlighting this. The scope of practice covers health promotion and disease prevention in LMICs where they are relatively  very valuable due to the growing burden of NCDs. However, lack of resources, research and capacity building are poor making it more challenging. The lack of literature is revealing of these challenges and factors and highlighted within Line 75-99 and Line 307-317.  

Line 314 in the paragraph commencing “three studies employed…” there is 20xx and 20yy – this needs to be addressed. 

Authors response: it has been addressed in line 343.

 It is interesting that the use of technology was so limited to telehealth (video-conferencing) and remote monitoring using a Fitbit, when there have been behaviour change apps being trialled for many years. The search strategy did not intentionally search from digital health intervention as it did not include terms like mHealth, eHealth, telehealth etc which may be why there were limited papers. Also, physiotherapists may not be leading digital interventions because it has not been a part of the entry-level curriculum and we are seeing a lag in uptake of digital strategies. 

Authors response: You are correct, a few papers that we found on digital health were not led by physiotherapists.

Comments on the Quality of English Language

Well written. 

As in previous section, recommend re-considering the use of so many abbreviations. There are some abbreviations that could be written in full with little impact on word count and it would improve the readability. 

Authors response: This was addressed with a few abbreviations maintained because of word count especially since more literature was added.

There are some missing dates and references throughout that need to be addressed. 

Authors response: Thank you for highlighting, attempts have been made to address all the corrections.

Reviewer 4 Report

Comments and Suggestions for Authors

Overall an interesting scoping review that looks at the scope of what topics do physiotherapy-led health promotion include when working among people with CVDs. Paper was well written and in-line with a scoping review guideline. I do have a couple of queries to raise with the authors regarding the paper. 

1. While the authors did briefly mentioned about why physiotherapists in LMICs often do need to do health promotion activities, I do think that the rationale for the scoping review needs to be stronger. Why are we focusing on physiotherapy-led health promotion? How will this differ from a standard health promotion campaign that is delivered by other health professionals or someone from public health? Another couple of sentences in the introduction to highlight the rationale as to why it needs to be a "physiotherapy-led health promotion" effort may be important. Perhaps something around our scope of practice and how we do have a responsibility in preventative health, even though the vast majority of time is spent in acute and rehabilitation settings. 

2. To aid the presentation for Table 3, may I suggest that authors consider dividing the table into two sections. One section is more about the intervention that physiotherapists promoted while the second section was about how the health promotion intervention was conducted by the physiotherapist. I think it is important to separate the two out and then you can report on them accordingly as I believed that all studies would have reported on the components and how they went about delivering it. 

3. Discussion could be strengthened by discussing about the scope of practice of a physiotherapist in a little more detail. The scope of practice issue was just skimmed past in the 5th paragraph when I would have thought that was the main point that needed to be highlighted. While authors did highlight that things like sleep, smoking cessation and alcohol abuse are within scope of physiotherapist, I think the authors can flashed this out a little further about what they meant by adequate training. How much is enough? Who should be delivering such training? In a context that we often work in a multidisciplinary team setting, would it be better for the physiotherapist to refer on to others rather than trying to deliver the health promotion on their own? 

4. It could be potentially a limitation that the review focused on physiotherapist-led health promotion interventions that appears to be structured. As a physiotherapist, we would often provide patient with advice or education about healthy lifestyle habits but this may not be captured in a structured program. Therefore, potentially limiting the generalisability of the findings. 

Author Response

Overall an interesting scoping review that looks at the scope of what topics do physiotherapy-led health promotion include when working among people with CVDs. Paper was well written and in-line with a scoping review guideline. I do have a couple of queries to raise with the authors regarding the paper. 

  1. While the authors did briefly mentioned about why physiotherapists in LMICs often do need to do health promotion activities, I do think that the rationale for the scoping review needs to be stronger. Why are we focusing on physiotherapy-led health promotion? How will this differ from a standard health promotion campaign that is delivered by other health professionals or someone from public health? Another couple of sentences in the introduction to highlight the rationale as to why it needs to be a "physiotherapy-led health promotion" effort may be important. Perhaps something around our scope of practice and how we do have a responsibility in preventative health, even though the vast majority of time is spent in acute and rehabilitation settings. 

Authors response: Attempt to strengthen the rationale for the scoping review have been made in the fourth paragraph line 81-95

  1. To aid the presentation for Table 3, may I suggest that authors consider dividing the table into two sections. One section is more about the intervention that physiotherapists promoted while the second section was about how the health promotion intervention was conducted by the physiotherapist. I think it is important to separate the two out and then you can report on them accordingly as I believed that all studies would have reported on the components and how they went about delivering it. 

Authors response: Table 3 has been formatted to improve readability Page 9.

  1. Discussion could be strengthened by discussing about the scope of practice of a physiotherapist in a little more detail. The scope of practice issue was just skimmed past in the 5th paragraph when I would have thought that was the main point that needed to be highlighted. While authors did highlight that things like sleep, smoking cessation and alcohol abuse are within scope of physiotherapist, I think the authors can flashed this out a little further about what they meant by adequate training. How much is enough? Who should be delivering such training? In a context that we often work in a multidisciplinary team setting, would it be better for the physiotherapist to refer on to others rather than trying to deliver the health promotion on their own? 

Authors response: Discussion on physiotherapy in the assessment of lifestyle and behaviour change have been improved line 325 -339.

  1. It could be potentially a limitation that the review focused on physiotherapist-led health promotion interventions that appears to be structured. As a physiotherapist, we would often provide patient with advice or education about healthy lifestyle habits, but this may not be captured in a structured program. Therefore, potentially limiting the generalisability of the findings. 

Authors response: Thank you, this is so true as physiotherapy health promotion tends to be opportunistic, not like in a fully organised cardiac rehabilitation program, though we receive patients that need the health promotion. The focus usually tends to be on curative or therapeutic interventions. Whilst interventions tend to be opportunistic, we can understand the range of likely opportunistic interventions that physiotherapists need to be familiar with from structured programmes of care.

Round 2

Reviewer 4 Report

Comments and Suggestions for Authors

Thank you for taking the time to consider the suggestions and draft the responses. The responses have been addressed and I am happy with the current state of the manuscript.